# Nature of Charge Transfer Effects in Complexes of Dopamine Derivatives Adsorbed on Graphene-Type Nanostructures

**DOI:** 10.3390/ijms251910522

**Published:** 2024-09-29

**Authors:** Alex-Adrian Farcaş, Attila Bende

**Affiliations:** National Institute for Research and Development of Isotopic and Molecular Technologies, Donat Street, No. 67–103, RO-400283 Cluj-Napoca, Romania; alexfadrian@yahoo.com

**Keywords:** TDDFT, charge transfer, graphene, dopamine, zwitterion, quinone

## Abstract

Continuing the investigation started for dopamine (DA) and dopamine-o-quinone (DoQ) (see, the light absorption and charge transfer properties of the dopamine zwitterion (called dopamine-semiquinone or DsQ) adsorbed on the graphene nanoparticle surface is investigated using the ground state and linear-response time-dependent density functional theories, considering the ωB97X-D3BJ/def2-TZVPP level of theory. In terms of the strength of molecular adsorption on the surface, the DsQ form has 50% higher binding energy than that found in our previous work for the DA or DoQ cases (−20.24 kcal/mol vs. −30.41 kcal/mol). The results obtained for electronically excited states and UV-Vis absorption spectra show that the photochemical behavior of DsQ is more similar to DA than that observed for DoQ. Of the three systems analyzed, the DsQ-based complex shows the most active charge transfer (CT) phenomenon, both in terms of the number of CT-like states and the amount of charge transferred. Of the first thirty electronically excited states computed for the DsQ case, eleven are purely of the CT type, and nine are mixed CT and localized (or Frenkel) excitations. By varying the adsorption distance between the molecule and the surface vertically, the amount of charge transfer obtained for DA decreases significantly as the distance increases: for DoQ it remains stable, for DsQ there are states for which little change is observed, and for others, there is a significant change. Furthermore, the mechanistic compilation of the electron orbital diagrams of the individual components cannot describe in detail the nature of the excitations inside the complex.

## 1. Introduction

Charge transfer (CT) effects govern the functional principle of many complex physical phenomena found in photovoltaic [1,2,3,4,5], photocatalytic [6,7,8] or other optoelectronic materials [9,10]. The basic concept of the CT phenomenon was first introduced by Mulliken [11,12,13], but since then, the physical model has progressed considerably, as it has become increasingly important in the development of materials with special properties [14,15,16,17].

In general, the theoretical description of CT effects can be formulated based on the first-order rate equation in the framework of Marcus theory [18,19,20]. However, in order to understand the global picture of CT effects in biological, molecular electronics, and photonics processes, it is important to assume the so-called multifaceted nature of the charge transfer phenomenon [21,22,23,24,25]. One of these special cases is when the CT effects are induced through the photon absorption by one of the constituents of the system, and either a photoinduced electron transfer (PET) or a hole transfer (PHT) is generated [22]. The resulting photoelectron (or hole) is then the trigger for phenomena such as photocatalysis [26,27,28], photosensitization [29,30], or photovoltaics [5,31].

Energy transfer and its dynamics based on charge carriers at the molecule–semiconductor interface play a fundamental role in almost all the elementary steps required for surface chemistry [32]. Recently, in a previous study [33], we showed that the light-induced charge transfer effects between the graphene nanoparticle (GrNP) surface and the dopamine (DA) molecule adsorbed on it depend on the oxidative state of dopamine (catechol or o-quinone forms). Namely, the charge transfer excitations were characterized as molecule–surface charge transfer (or MSCT) for the GrNP–DA complex and as the reverse, i.e., surface-to-molecule charge transfer (or SMCT) for the GrNP–dopamine-o-quinone (or DoQ) complex. The difference between the two cases is due to the presence of an energetically low-lying unoccupied orbital (LUMO+1 of the complex) in the case of DoQ, which allows charge transfer from the surface to the molecule. This means that the nature of the electron orbitals formed within a complex system is crucial for the charge transfer between subsystems. On the other hand, the photochemical behavior of the isolated dopamine is strongly dependent on its concentration and tautomer forms as well as on the nature and the pH of the solvent environment [34,35]. Moreover, DA through the self-polymerization process can form thin, surface-adherent polydopamine films (PDAs) [36,37], where different tautomers or quinone forms of DA build the polymer hetero structure chain. In this regard, it is worth studying the surface-adsorption and light-absorption properties of not only the two extremes of the oxidation process [38] but also those of the intermediates, such as the protonated dopamine-semiquinone (DsQ), called the DA-zwitterion [34] (see Figure 1c).

On the other hand, the 2D graphene as well as its finite-sized graphene nanoparticle show intriguing electronic structure properties [39,40,41,42,43,44,45]. Accordingly, the characteristic interaction between graphene and dopamine has been explored primarily to improve dopamine sensing and detection techniques [46,47,48]; however, the optical spectroscopic characteristics of this cluster association have been less studied.

The aim of our study is to give a detailed description of the nature of electron transitions induced by electromagnetic radiation in the case of graphene–DsQ and of the nature of charge transfer as a function of the distance between the molecule and the surface in all three cases (GrNP–DA, GrNP–DsQ and GrNP–DoQ).

## 2. Results and Discussion

### 2.1. DsQ Adsorbed on the GrNP Surface

#### 2.1.1. Structure

The fully aromatic hydrocarbon structure containing three zig-zag and seven armchair edges as a model system for the graphene nanoparticle (GrNP) is considered (see the complex shown with “tube” type representation in Figure 2). This system can be considered a zero-dimensional nanoparticle with a large enough size to describe the interaction of the graphene surface with the molecule to a good approximation, especially as far as the inner core of the honeycomb is concerned. Its geometry is fully optimized considering the ωB97X-D3BJ/def2-TZVPP/CPCM (water) level of theory (for details, see Section 3). The geometry optimization leads to an irregular honeycomb lattice for GrNP, where the C–C bond length varies between 1.38 and 1.47 Å. Hereafter, the geometry of the GrNP is kept fixed, on which the DsQ molecule is superimposed, and the relative position of the molecule with respect to the GrNP is optimized using the same theoretical method. The geometry configuration of the DsQ adsorbed on the GrNP surface can be seen in Figure 2. In the equilibrium geometry configuration of the binary complex, the DsQ lies almost parallel to the graphene sheet; the distances between the graphene plane and the heavy atoms of the catechol fragment are between 3.369 and 3.463 Å; and the N atom of the amine group is at 3.63 Å far from the plane (see Figure 2b). The adsorption energy, defined as the difference between the energy of the binary complex and the sum of the energies of the individual fragments [49], is −30.41 kcal/mol, which is more than 50% higher than the adsorption energies obtained for DA (−20.24 kcal/mol) and DoQ (−20.11 kcal/mol), respectively (see Ref. [33]). This also means that DsQ adheres better to the graphene surface than either DA or DoQ molecules. As for the ground state charge distribution, there is a charge transfer from DsQ to the surface of 0.070 *e* (so-called molecular–surface charge transfer or MSCT), which is very close to the value obtained for DA (0.068 *e*, see Ref. [33]). As for the difference in conformational energy between DA and DsQ adsorbed on graphene, it is found that the GrNP–DA complex is more stable than GrNP–DsQ with an energy value of 10.55 kcal/mol. However, in a very recent study, it was shown that the PCM model incorrectly prefers the neutral form to the zwitterion [50], which may, for example, give a discrepancy between DA and DsQ results in the present case. In this respect, e.g., the CANDLE solvent model could give more accurate results [50,51].

#### 2.1.2. UV Absorption

In order to better understand the UV absorption spectrum of the binary system and the fingerprints of the different components, the first thirty electronically excited states (S_i_, i = 1, …, 30) are calculated both for the binary system and the components separately (see the black, red and blue curves in Figure 3). Taking into account that the results for DsQ have already been reported (see results for dopamine zwitterion (or DA^*zw*^) presented in Ref. [34]), only a brief summary of its photochemical behavior is given. In the present case, a slightly different result is obtained since instead of the DLPNO-STEOM-CCSD method developed at the coupled-cluster theory level, the calculations are performed with TDDFT. Accordingly, DsQ presents spectral characteristics with peak maxima at 259 nm (S_1_) and 224 nm (S_2_). See the blue spectral curve in Figure 3. The one-electron molecular as well as the natural difference orbitals involved in the electronic excitations of the DsQ molecule can be found in the Appendix A.

The geometry optimization of the first excited electron state (S_1_) leads to a shortening of the C–O bonds (from 1.371 Å to 1.367 Å and from 1.298 Å to 1.270 Å) and to a stretching of the carbon ring, with all six bonds being longer than 1.4 Å (the bond values are 1.427, 1.456, 1.400, 1.414, 1.400, and 1.424, respectively, in Å). The fluorescence emission wavelength is 308 nm.

Graphene is a 2D solid that has interesting physics due to its unusual electronic band structure [39,40,43,44]. Of course, this is accompanied by unusual optical properties characterized by a broad absorption spectral range, given by intra-band transitions (π–π*) at low photon energies (0.0–5.0 eV in the far-infrared spectral range) and inter-band transitions (π–σ*) at higher energies (more than 5.0 eV from the mid-infrared to the ultraviolet) [52,53]. As described in more detail in Section 3, the simplified zero-dimensional (finite sized) GrNP model is able to reproduce most of the spectral features observed in the 2D model. The UV absorption spectrum for GrNP is shown with red line in Figure 3. In this case, two characteristic peaks are observed, one at 566 nm and the other at 270 nm. The first one appears as an effect of a GrNP finite dimensional model, while the second one is the absorption maximum typical of a 2D graphene sheet [33,53].

The UV absorption spectrum of the GrNP–DsQ complex is shown in black in the same Figure 3, while the electronic excited state energies, oscillator strengths, amount of charge transfers between the subsystems and the nature of the electronic states (either localized on one of the subsystems or CT state) for the first thirty electronic excited states are collected in Table 1. For the spectrum of the binary system, it is observed that both peaks seen at 566 nm and 270 nm for the pristine GrNP system are almost identically preserved. A very small intensity increase is observed for the ground to second excited state (S_0_ → S_2_) electronic transition (λ(S_2_) = 471 nm). As for the nature of the excited states, ten are localized on the GrNP, and eleven are of the CT type, while nine are mixed, i.e., containing both localized and CT-type excitation (see data in Table 1, the natural difference orbitals (NDOs) collected in Appendix A and the natural transition orbitals (NTOs) collected in Appendix A). Compared to the GrNP–DA and GrNP–DoQ cases, where seventeen and eighteen electron excitations are localized on GrNP, respectively, DsQ is characterized by fewer localized and more CT-type electron transitions, with no pure DsQ-like excitations. Interestingly, it is observed that excitations localized on GrNP have lower (with the exception of S_0_ → S_2_), while pure CT-type excitations are characterized by higher energy (higher than S_0_ → S_12_) transitions. The strongest CT nature of the excited electron states for DsQ is also confirmed by the amount of charge transferred between the two subsystems. While in the case of GrNP–DA, there are only three cases where the transferred charge is greater than 0.3 *e* and only one case greater than 0.7 *e*, for the present system, there are eleven states with a CT greater than 0.3 *e*, three cases with a CT greater than 0.7 *e*, and for the S_2_ state, even one unitary charge is transferred from the DsQ to the graphene nanoparticle (called molecule-to-surface charge transfer or MSCT effects). To understand the reason for this significant charge transfer, it is necessary to see what the nature of the highest occupied and lowest unoccupied electron orbitals are. Accordingly, based on the fragment orbital contribution analysis, the molecular orbital scheme of the joined GrNP–DsQ system is built. Its graphics can be seen in Figure 4. By comparison with the GrNP–DA case, it can be observed that since the HOMO (or Highest Occupied Molecular Orbital) energy of the individual DsQ molecule is higher than that of the DA and thus approaches the HOMO energy of GrNP, not only will the HOMO-1 orbital of the GrNP–DsQ complex contain a DsQ contribution (more than 80% DsQ) but also the HOMO (18% DsQ). And, of course, not only the HOMO and HOMO-1 orbitals contain smaller or larger DsQ contributions but also the other lower-lying occupied orbitals (see the molecular orbitals collected in Appendix A). As for the unoccupied orbitals, the first eleven are almost entirely localized on GrNP, but also the subsequent three orbitals contain only a small proportion of DsQ contributions (Figure 4).

Accordingly, in the molecular orbital excitation scheme, the lowest electronic excitation partially localized on DsQ (S_0_ → S_25_) involves the LUMO + 15 unoccupied orbitals (see Appendix A), where LUMO means the Lowest Unoccupied Molecular Orbital. Overall, it can be concluded that the nature of the electronic transitions observed in GrNP–DsQ is very similar to that seen in GrNP–DA, with the addition that in the first case, charge transfer electronic excitations appear to be more pronounced.

#### 2.1.3. Excited State Relaxation

From the analyses carried out so far, it can be concluded that excited states can be localized on one or the other component of the molecule–surface complex, or charges can easily migrate from the molecule to the surface, or vice versa [33]. It can therefore be assumed that relaxation processes for different electronic excited states can occur either on the components separately or even inside the binary system. It has already been shown that non-radiative relaxation can occur either on the graphene surface itself [54] or in the binary complex [55,56,57,58,59,60], while excited states can also decay via fluorescence phenomena in the case of slightly modified GrNP [61,62]. Furthermore, it is also shown that for the S_1_ excited state, there is no significant geometric change for DA adsorbed on GrNP, whereas for DoQ, significant changes appear for both C=O and aromatic C–C bonds due to excitation [33]. To see exactly what effect the excitation has on the DsQ molecule adsorbed on GrNP, the geometry of the molecule is optimized in the first excited state. Similar approaches to those for DA and DoQ are considered in the present case. Namely, (i) non-radiative phenomena are neglected due to the very complex theoretical framework and huge computation resources; (ii) the positions of the carbon atoms forming the surface are frozen. When comparing the ground and first excited state geometries, it can be observed that there is a slightly more pronounced change in the molecular bonds than is seen for DA but not nearly as significant as that obtained for DoQ. Accordingly, the C–O^−^ and C–OH bonds stretch from 1.283 Å and 1.359 Å to 1.302 Å and 1.372 Å, respectively, while the aromatic C–C bond lengths oscillate from 1.411–1.383–1.395–1.398–1.374–1.429 Å to 1.405–1.397–1.388–1.404–1.377–1.426 Å. Regarding the relative position of the DsQ, the plane defined by the aromatic ring fragment comes closer to the GrNP surface and appears to be tilted, with the six carbon atoms between 3.08 and 3.27 Å away from the plane of the surface, the O atoms at 3.00 Å and 3.04 Å, respectively, as well as the N atom of the amine group at 3.36 Å. At the same time, the wavelength of the S_0_ → S_1_ excitation changes from 570 nm to 627 nm. In conclusion, it can be stated that no significant geometric change occurs in the DsQ molecule during the relaxation of the S_1_ excited state, unlike in DoQ (see Ref. [33]).

### 2.2. Charge Transfer and Distance from Surface

The effects of changing the separation distance of the charge transfer interactions between luminescent QD and proximal dopamine (in QD–dopamine assemblies) has been already demonstrated by steady-state and time-resolved fluorescence measurements considering different lengths of poly(ethylene glycol) (PEG) as a bridge [63]. In light of this, the question arises as to what extent the perpendicular shift between the plane of the DA, DoQ, and DsQ molecules and the GrNP surface affects the charge transfer phenomenon in these assemblies. Accordingly, the molecules are moved along the z-coordinate axis by different values (Δ*z* = −0.3, −0.2, −0.1, 0.1, 0.2, 0.3, 0.4, 0.5, 0.6, 0.8 and 1.0 Å) relative to the equilibrium distance (*z_0_*) from the surface. Separately for each of these geometries, the Löwdin electron population is calculated for the first thirty electronic excited states, based on the electron density of the given state. The amounts of charge transfer calculated for different plane distances relative to the equilibrium geometry for the DA, DoQ, and DsQ adsorbed on the GrNP surface are presented in Figure 5. Molecular orbital energy schemes (in eV) built based on the fragment orbital contribution analysis of the individual, GrNP, DA, DoQ, and DsQ components and of the mixed GrNP–DA, GrNP–DoQ, and GrNP–DsQ binary complexes computed for Δ*z* = −0.3, 0.0 and +1.0 relative stacking distances values are shown in Figure 6.

#### 2.2.1. DA

It has already been shown that the nature of the charge transfer between the DA and the GrNP substrate is MSCT (or metal-to-surface charge transfer) [33]. For the present case, at the equilibrium geometry configuration (*z*_0_ + Δ*z*, where Δ*z* = 0.0), it can be observed that there are three distinct sets of electronic excited states with significant CT character (see Figure 5a). The first set is formed by S_6_, S_7_ and S_9_, and the second group is formed by S_15_, S_16_ and S_17_, while the third set is defined by S_25_–S_29_ states. When the DA plane approaches the surface (Δ*z* = −0.3, −0.2, and −0.1 Å), it is observed that the number of CT-like electronically excited states increases, but the charge transfer amount of the existing CT states decreases and that of the newly created CT states increases. For Δ*z* = −0.3, only S_1_, S_3_, and S_5_ show a charge transfer lower than 0.1 *e*, but for S_6_, for example, the 0.4 emphe CT obtained at Δ*z* = 0.0 Å for −0.3 Å decreases to 0.18 *e*. Furthermore, a shift in the CT peaks also can be observed as the distance between the planes decreases. For example, the peak of S_7_ at Δ*z* = 0.0 Å becomes S_6_ at −0.3 Å, S_16_ becomes S_15_, and S_26_ mainly bifurcates into S_25_ and S_27_. As the distance between the planes increases, the opposite effect is observed. That is, the number of CT states decreases, but the amount of charge transfer increases. The first set of excited states (S_6_, S_7_ and S_9_) collapses as S_14_ for Δ*z* = +1.0 Å, and the second group as S_26_, while the third group moves out of the S_1_–S_30_ interval proposed for the present analysis. However, for S_14_ and S_26_, the amount of charge transfer is 0.87 *e* and 0.95 *e*, respectively. Fragment orbital contribution analysis reveals that the amount of this charge transfer as a function of Δ*z* depends on the degree of overlap of the individual molecular orbitals. Indeed, for Δ*z* = −0.3, the HOMO and HOMO-1 orbitals contain larger DA-type contribution, which means stronger CT effects between the DA and the GrNP substrate, while in the case of Δ*z* = +1.0, there are hardly any mixed molecular orbitals, and therefore the number of CT-type excited electron states is significantly reduced. In the latter case, the S_12_ electron excitation is characterized by HOMO-2 → LUMO, while S_26_ is characterized by HOMO-8 → LUMO and HOMO-9 –> LUMO one-electron transitions, respectively (see Figure 6a).

#### 2.2.2. DoQ

For the CT states of the complex formed by the DoQ molecule in adsorption on the GrNP surface, the charge is mostly transferred from the surface to the molecule (called SMCT or surface-to-molecule charge transfer) [33], in contrast to the previous DA case. At the equilibrium geometry position, for Δ*z* = 0.0 Å, also three distinct sets of electronic excited states with significant CT character can be observed (see Figure 5b). The first set is made up of the S_3_ state alone, the second is formed by the S_16_, S_18_ and S_21_ states, and the third is the S_30_ state. In the decreasing distance direction between the planes (Δ*z* = −0.3, −0.2 and −0.1 Å), no substantial change in the charge transfer phenomenon is observed. Rather, the transferred charge is slightly dispersed in other states (see S_9_–S_15_), but the position of states with high CT values in the excitation spectrum does not change. A similar trend can be observed for the increasing distance direction between the planes. The S_3_ excited state still behaves as a CT state even at Δ*z* = +1.0 Å. The set of S_16_–S_21_ is slightly rearranged. Here, states S_20_ and S_23_ will show the strongest charge transfer character, while state S_30_ moves out from the interval proposed for the present analysis. Basically, it can be said that for DoQ–GrNP, the CT phenomenon does not show any significant change in the distance interval of −0.3 Å and +0.1 Å. This can also be explained by the fact that, based on the fragment orbital contribution analysis, LUMO + 1 is essentially a DoQ-type molecular orbital, and by varying the distance (Δ*z*), the orbital nature almost remains the same (see Figure 6b). For example, the CT-type S_3_ state is defined by the HOMO → LUMO + 1 one-electron transitions.

#### 2.2.3. DsQ

It has already been shown in the previous Section 2.1 that the photochemical behavior of DsQ is similar to that of DA since the nature of the CT effects is the same MSCT-type transition as that observed for DA. The difference is that the number of pure CT-type transitions or their mixing with localized GrNP-type excitation within the first thirty excited states is significantly higher for DsQ than for DA, and on average, the magnitude of the charge transfer is also higher (compare Figure 5a,c). At the equilibrium geometry position of the DsQ, i.e., with Δ*z* = 0.0 Å, significant CT transitions are seen for the S_2_ state and for all other excited states starting from S_13_. For a molecular configuration approaching the surface, e.g., Δ*z* = −0.3 Å, the intensity of charge transfer increases, and the CT nature is already observed at S_11_. When the molecule starts to move away from the surface, the opposite happens, i.e., intensities decrease and, for example, at Δ*z* = +0.4 Å the S_13_–S_15_ states hardly contain any CT nature. At Δ*z* = +1.0 Å only S_2_, S_16_, and S_30_, and partially the groups of S_20_–S_27_, show CT character. Similar to DA, the CT transitions for DsQ are strongly dependent on the distance between the molecule and the surface, but the number of CT states is still larger than that observed for DA. Compared to DA, the change is that for Δ*z* = +1.0 Å, the DsQ-type orbital is no longer HOMO-2, but HOMO-1 with lower orbital energy (see Figure 6c).

## 3. Materials and Methods

The equilibrium geometries are computed in the framework of density functional theory (DFT) considering the ωB97X-D3BJ exchange–correlation (XC) functional [64,65] by including Grimme’s empirical dispersion correction scheme [66,67] and using the def2-TZVPP triple-ζ basis set of the Karlsruhe group [68] as implemented in the Orca program suite [69,70]. Based on the large benchmark calculation, ωB97X-D3BJ is proved to be one of the most performant XC functionals among the range-separated hybrid GGA (generalized gradient approximation) functionals [71]. The water solvent environment is taken into account through the conductor-like polarizable continuum (CPCM) model [72,73]. The electronically excited state calculations are performed using the time-dependent version of the same DFT framework considering the Tamm–Dancoff approximation (TDA) [74]. The RIJCOSX approximation [75,76,77,78,79] designed to accelerate Hartree–Fock and hybrid DFT calculations are considered together with the Def2/J [80] auxiliary basis set for Coulomb fitting and the def2-TZVPP/C [81] auxiliary basis set for correlation fitting in the case of TD-DFT calculations. The amount of transferred charge between the complex constituents is obtained based on the analysis of the charge population (Löwdin atomic charges) of a given electronically excited state density computed in the TDDFT framework. The molecular geometries are built, analyzed and further manipulated using the Multiwfn 3.7 [82], Gabedit 2.5.2 [83], and Avogadro 1.2.0 [84] programs, while the molecular graphics are created using the GaussView 6.1 [85] software. For the GrNP model, a two-dimensional (three zig-zag and seven armchair edges) rectangular graphene nano-sheet with fully aromatic hexagonal rings is considered. Most features of the experimental UV absorption spectra of GrNPs [52,53,86] can already be theoretically reproduced by the TDDFT method considering the above-mentioned 0-dimensional (finite sized) aromatic hydrocarbon structure model [33]. This simplified surface model with its discrete electron states can act as an environment for interacting with adsorbed molecules, where these electron states can combine appropriately with the molecule’s electron states and realistically reproduce the electron excitation behavior of the molecule on the surface. Of course, this model tends to describe only the electron states of the adsorbed molecule more accurately; if one wants to follow the changes in the electronic band structure of graphene, the simplified model described here is no longer suitable, and a solid-state model is needed [87,88].

## 4. Conclusions

In the present work, the equilibrium geometry structure and light absorption properties of the dopamine-semiquinone (DsQ) adsorbed on the graphene surface is computed using the ground state and linear-response time-dependent density functional theories at the ωB97X-D3BJ/def2-TZVPP level, and the results are compared with those obtained previously for the dopamine (DA) and dopamine-o-quinone (DoQ) [33]. Analyzing the adsorption energies between the individual molecules and the graphene substrate, it is shown that the DsQ form has 50% higher binding energy than that previously found for the DA or DoQ cases (−30.41 kcal/mol vs. −20.24 kcal/mol and −20.11 kcal/mol). Furthermore, analyzing the natural difference orbital profiles characteristic for the vertical electronic excitation, three different types of electronic transitions are identified, i.e., locally excited on graphene and locally excited on the molecule and charge transfer (CT) states characteristic of the charge migration induced by the excitation. In terms of the nature of charge transfer, DA- and DsQ-based complexes show a high degree of similarity, i.e., in both cases, the transfer is from the molecule to the surface, whereas for DoQ, the transfer direction is the opposite. As for the comparison between DA- and DsQ-based complexes, in the latter case, the number of CT-type electronic transitions and the amount of charge transfer is significantly higher. By varying vertically the adsorption distance between the molecule and the surface, the amount of charge transfer obtained for DA and DsQ decreases significantly as the distance increases, while for DoQ, this dependence is much weaker. Based on these findings, it can be emphasized that a mechanistic compilation of the electron orbital diagrams of the individual components (for example, see Figure 1 in [4] or Figure 4 in [5]) cannot describe in detail the nature of the excitations inside the complex since the electronic excitations occur directly between different mixed molecular–surface one-electron orbitals. The mixture of these orbitals depends strongly on the degree of overlap of the individual components, which in turn is determined by the distance between the components. In general, for the three complexes studied, the number of CT-type electron transitions, the amount of charge transferred, and the direction of transfer between the molecule and the substrate are different from one dopamine derivative to another. In this way, if one can control the protonation state of dopamine units, it is possible to influence charge transfer phenomena between the dopamine polymer and the graphene substrate.

## Figures and Tables

**Figure 1 ijms-25-10522-f001:**
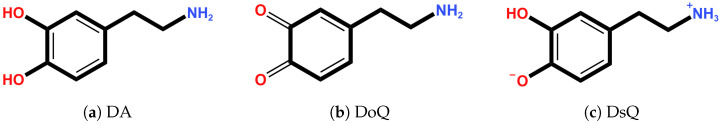
The 2D chemical structure of (**a**) dopamine (or DA), (**b**) dopamine-o-quinone (or DoQ) and (**c**) dopamine-semiquinone (or DsQ).

**Figure 2 ijms-25-10522-f002:**
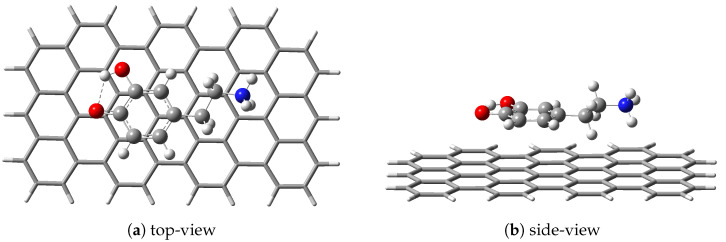
The geometry configuration of the dopamine-semiquinone (or DsQ) adsorbed on the GrNP surface computed at ωB97X-D3BJ/def2-TZVPP/CPCM level of theory.

**Figure 3 ijms-25-10522-f003:**
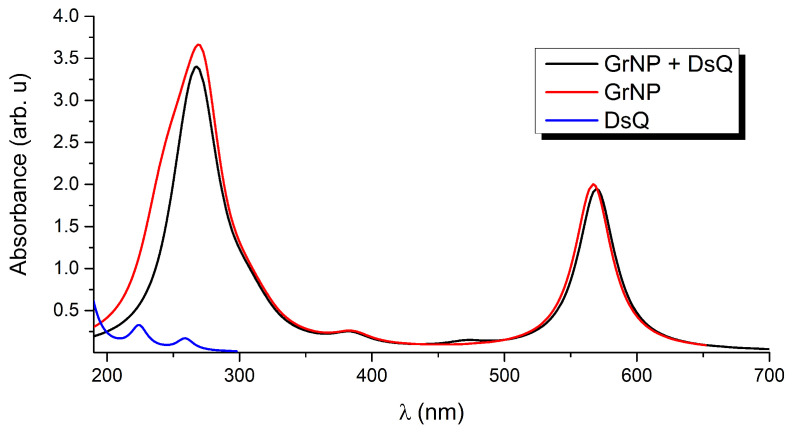
Theoretical UV absorption spectra of the graphene–DsQ binary complex and the individual constituents computed at the TDDFT/ωB97X-D3BJ/def2-TZVPP/CPCM level of theory.

**Figure 4 ijms-25-10522-f004:**
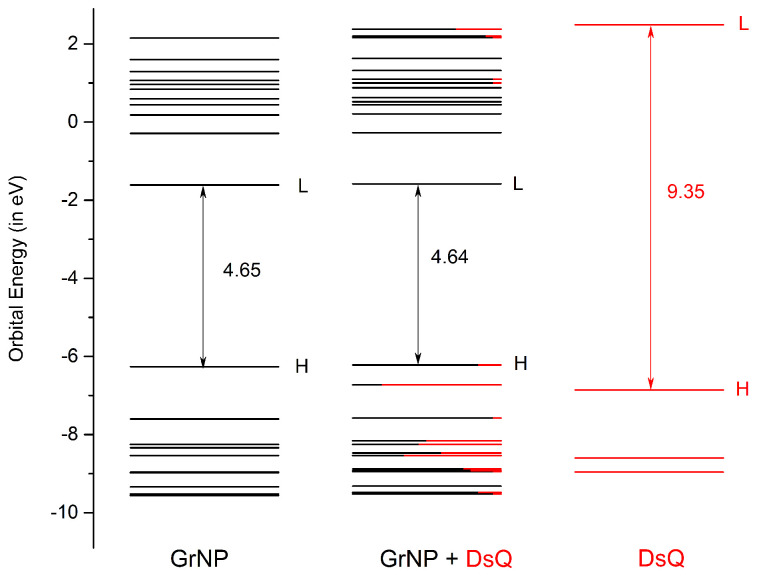
The molecular orbital energy scheme (in eV) of the individual, GrNP, and DsQ components and of the mixed GrNP–DsQ binary complex (H = HOMO (or Highest Occupied Molecular Orbital), L = LUMO (or Lowest Unoccupied Molecular Orbital)) based on the fragment orbital contribution analysis.

**Figure 5 ijms-25-10522-f005:**
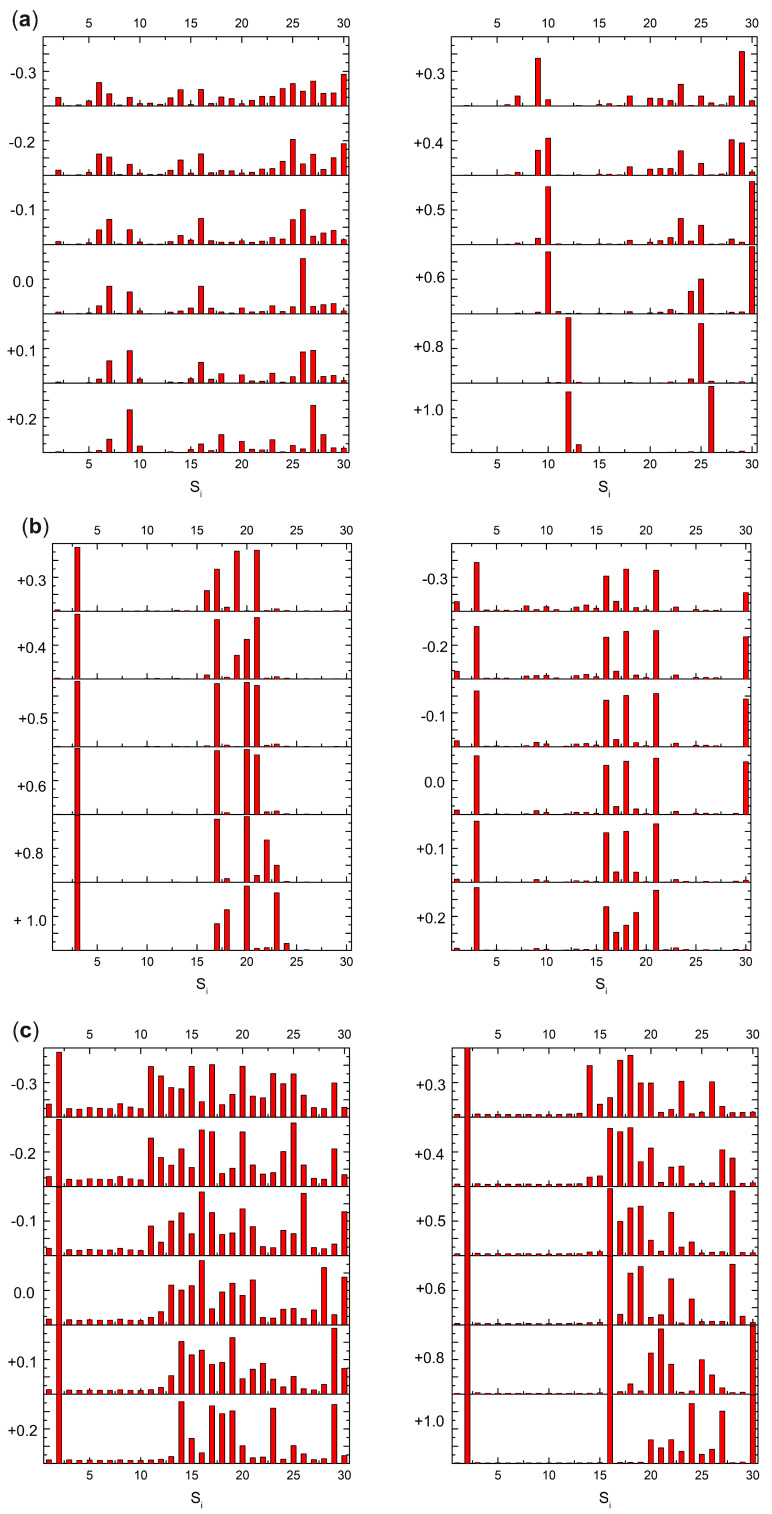
The transferred charge (between 0 and 1 values of the elementary charge) calculated for different plane distances relative to the equilibrium geometry (z_0_ + Δz, Δz between −0.3 Å and +1.0 Å) for DA (**a**), DoQ (**b**) and DsQ (**c**), respectively, adsorbed on the GrNP surface, computed at the TDDFT/ωB97X-D3BJ/def2-TZVPP level of theory.

**Figure 6 ijms-25-10522-f006:**
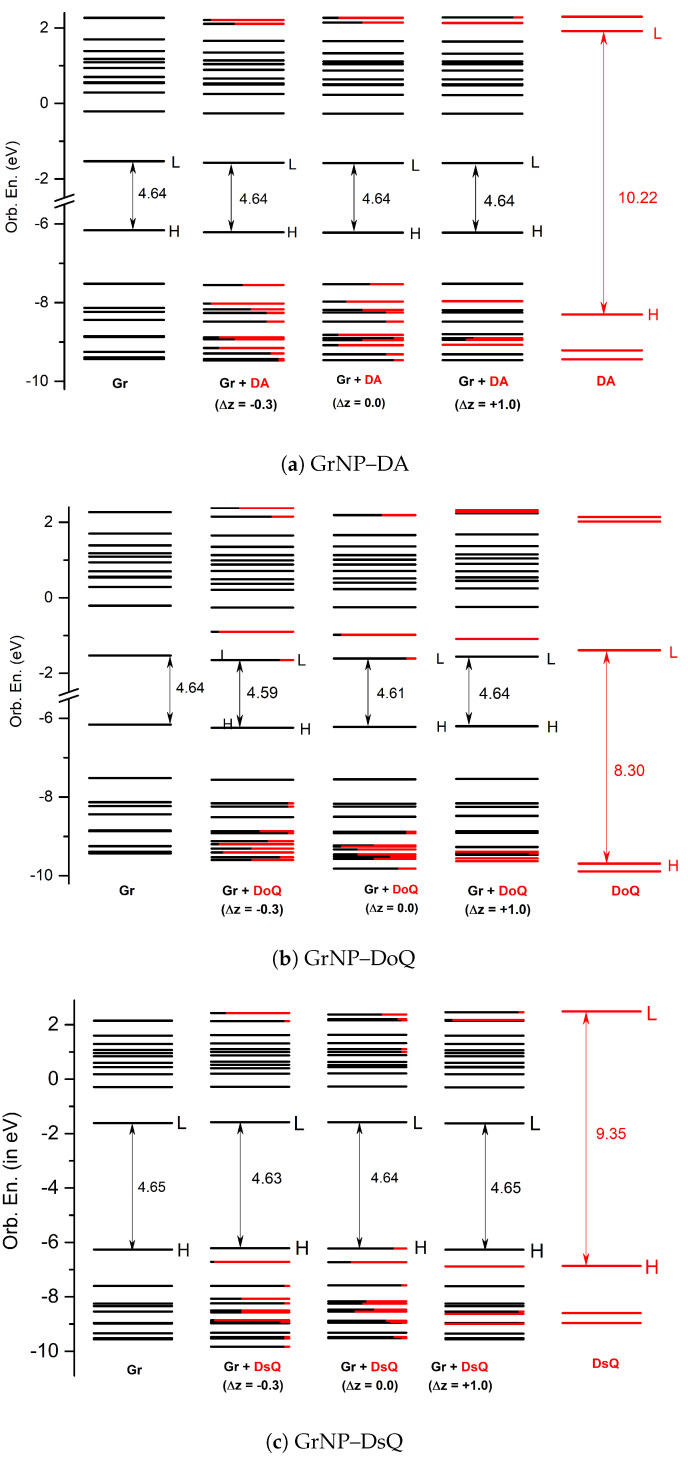
Molecular orbital energy schemes (in eV) built based on the fragment orbital contribution analysis of the individual, GrNP (1st col.), DA, DoQ, and DsQ (last col.) components and of the mixed (**a**) GrNP–DA, (**b**) GrNP–DoQ, and (**c**) GrNP–DsQ binary complexes (H = HOMO, L = LUMO) computed for Δ*z* = −0.3 (2nd col.), 0.0 (3rd col.), and +1.0 (4th col.) relative stacking distances.

**Table 1 ijms-25-10522-t001:** The first thirty electronic excited state energies (in nm), their oscillator strengths, transferred charges, and electronic transition types (either located on the GrNP and on DsQ fragments or charge transfer states between the fragments) of the GrNP—DsQ binary complex, computed at the TDDFT/ωB97X-D3BJ/def2-TZVPP level of theory.

S*_i_*	1	2	3	4	5	6	7	8	9	10
λ (nm)	570	471	384	375	364	357	340	325	322	314
Osc. Str.	1.9323	0.0573	0.1312	0.0004	0.0000	0.0025	0.0023	0.0035	0.0144	0.1213
Charge tr. (*e*)	0.085	1.000	0.075	0.066	0.074	0.070	0.068	0.083	0.069	0.064
Type	GrNP	CT	GrNP	GrNP	GrNP	GrNP	GrNP	GrNP	GrNP	GrNP
S*_i_*	**11**	**12**	**13**	**14**	**15**	**16**	**17**	**18**	**19**	**20**
λ (nm)	305	304	301	297	295	290	285	284	282	273
Osc. Str.	0.2109	0.0050	0.0313	0.0253	0.0410	0.0020	0.0406	0.0407	0.0819	0.9082
Charge tr. (*e*)	0.110	0.190	0.576	0.505	0.569	0.930	0.233	0.478	0.604	0.429
Type	GrNP + CT	GrNT + CT	CT	CT	CT	CT	GrNP + CT	CT	CT	CT
S*_i_*	**21**	**22**	**23**	**24**	**25**	**26**	**27**	**28**	**29**	**30**
λ (nm)	270	268	266	262	261	260	259	258	256	254
Osc. Str.	0.6146	0.2963	0.5778	0.1547	0.4897	0.0171	0.0570	0.0121	0.2461	0.1927
Charge tr. (*e*)	0.651	0.107	0.099	0.227	0.237	0.095	0.215	0.828	0.148	0.690
Type	CT	GrNP + CT	GrNP + CT	GrNP + CT	DsQ + CT	GrNP	GrNP + CT	CT	GrNP + CT	CT

CT = Charge transfer character.

## Data Availability

Data are available on request.

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
