# Peer review of "Nature of Charge Transfer Effects in Complexes of Dopamine Derivatives Adsorbed on Graphene-Type Nanostructures"

_ijms, 2024, doi:10.3390/ijms251910522_

Round 1

Reviewer 1 Report

Comments and Suggestions for Authors

This is a theoretical article which deals with the charge transfer effects in complexes of dopamine derivatives adsorbed on graphene-type nanostructures. 

The manuscript needs to be revised and the following main points need to be carefully addressed before it can be considered for publication: 

1.     First, the novelty of the paper should be clearly evidenced in the manuscript, not only in the introduction, but also briefly in the abstract and in the conclusions. 

2.    It is mandatory that authors clarify in the manuscript the differences with their previous paper reported in Ref [33]. Some figures such as Fig. 2, 3, 4 and so on, together with Tab.1, have already been published in Ref. [33]. This cannot be acceptable for plagiarism. If the present manuscript is a continuation of the study in Ref. 33 it is necessary to remove all the text and figures already published and focus only on the new results, if any.

3.     The introduction needs to be improved. For sake of completeness, a wider overview of the potentiality of DFT calculation in the field should be given, such as its use for studying the first stage of formation of graphene, reporting the following remarkable and recent paper [https://doi.org/10.1016/j.carbon.2023.02.011]. Furthermore, an overview of other applications should be provided with some examples, one of which could be [https://doi.org/10.1021/acsanm.0c02307].

4.     Regarding the calculations reported in the manuscript, the authors should report whether there are experimental results in the literature for comparison with their calculations. 

5.     Furthermore, the conclusions should highlight in more detail the value that the manuscript adds to the current literature in the field and outline some perspectives opened by this study. 

Author Response

See our answers in the attached file.

Reviewer 2 Report

Comments and Suggestions for Authors

Using DFT calculations, the Authors study the adsorption strength, the UV-vis absorption and charge transfer for dopamine, dopamine-semi-quinone, and dopamine-o-quinone molecules adsorbed on small graphene nanosheets (graphene nanoparticles). They find that the dopamine-semi-quinone presents the highest adsorption strength and charge transfer. In order to explain the obtained results, the molecular orbital energy schemes of the complex of the graphene nanoparticle and the adsorbed molecule are computed. From these results, the Authors conclude that controlling the protonation state of dopamine could allow to modify the charge transfer between adsobed molecule and graphene nanoparticle.

The reported results are novel and could be interesting for specialists in this topic, but I feel that some points of the text need some clarification. Once this will be done, I suggest the publication of the manuscript.

1) line 10: since this is the first point in which "CT" is used, the Authors should indicate here that "CT" means "charge transfer"

2) line 20: remove "more"

3) line 33: change "than" into "then"

4) lines 61-62: even though it is clarified in Section 3, probably it would be better to specify here what you mean by "graphene nanoparticle"

5) lines 67-68 and 166: you are assuming that the presence of the molecule does not deform the geometry of the graphene nanoparticle; what do you think is the accuracy of this approximation?

6) lines 102-103: maybe here you could add some additional references about interesting physical properties of graphene; I suggest the following ones:

Katsnelson, M.; Novoselov, K.; Geim, A. Chiral tunnelling and the Klein paradox in graphene. Nature Phys 2006, 2, 620–625, DOI: 10.1038/nphys384

Macucci, M.; Marconcini, P. Theoretical Comparison between the Flicker Noise Behavior of Graphene and of Ordinary Semiconductors. J. Sensors 2020, 2020, 2850268, DOI: 10.1155/2020/2850268.

7) line 105: remove one of the brackets after "transitions"

8) lines 107 and 281: could you clarify what you mean by "0-dimensional GrNP model"?

9) line 114: add a comma between "energies" and "oscillator"

10) line 119: please clarify in the text if $S_0$ is the ground state

11) caption of figure 5: by "charge unit" do you mean $e$, the elementary charge?

12) lines 220-222 and 241-242: if I am not wrong, with $S_i$ you indicate states. Here you characterize a few states through some electron transitions; do you refer to transitions from the ground state to the considered excited states? This should be clarified in the text.

13) line 297: remove "with"

Author Response

See our answers in the attached file.

Reviewer 3 Report

Comments and Suggestions for Authors

In this work, the authors performed TDDFT calculations at the wB97X-D3BJ/def2-TZVPP level of theory to compute excited-state properties including excitation energies and oscillator strengths. The DsQ form has 50 % higher binding energy than that for the DA or DoQ cases (-20.24 kcal/mol vs. -30.41 kcal/mol). The results obtained for electronically excited states and UV-Vis absorption spectra show that the photochemical behavior of DsQ is more similar to DA than that observed for DoQ. The DsQ-based complex showed the most active CT phenomenon in terms of the number of CT-like states and the amount of charge transferred.

This work can be interesting for the computational chemistry and the industrial community. As such, the proposed article deserves to be published and the IJMS is certainly well targeted. Before the publication, I would like to ask the authors to consider the minor comments below.

1. page 4, Table 1

The unit for the oscillator strength is missing.

2. page 5, Figure 4

Are the orbital energies computed with DFT? It should be mentioned in the context that orbital energies computed at the DFT level have errors and the starting point dependence.

3. The abbreviations of “HOMO” and “LUMO” are not defined.

4. page 9, line 268

“framework considering the Tamm–Dancoff approximation (TDA) approximation”

Why TDA is used for TDDFT? TDA is mainly used in TDDFT to overcome the triplet instability, however, only singlet states are calculated in this work.

5. It is helpful to plot the natural transition orbitals obtained from TDDFT calculations.

Comments on the Quality of English Language

No major language or grammar problem found.

Author Response

See our answers in the attached file.

Round 2

Reviewer 1 Report

Comments and Suggestions for Authors

The authors have satisfactorily addressed all the raised issues, thus improving the manuscript. Therefore, the revised version of the manuscript is now suitable for publication.